# Fatty Acid Desaturase Involvement in Non-Alcoholic Fatty Liver Disease Rat Models: Oxidative Stress Versus Metalloproteinases

**DOI:** 10.3390/nu11040799

**Published:** 2019-04-08

**Authors:** Giuseppina Palladini, Laura G. Di Pasqua, Clarissa Berardo, Veronica Siciliano, Plinio Richelmi, Barbara Mannucci, Anna Cleta Croce, Vittoria Rizzo, Stefano Perlini, Mariapia Vairetti, Andrea Ferrigno

**Affiliations:** 1Department of Internal Medicine and Therapeutics, University of Pavia, 27100 Pavia, Italy; lauragiuseppin.dipasqua01@universitadipavia.it (L.G.D.P.); clarissa.berardo01@universitadipavia.it (C.B.); veronica.siciliano01@universitadipavia.it (V.S.); plinio.richelmi@unipv.it (P.R.); stefano.perlini@unipv.it (S.P.); andrea.ferrigno@unipv.it (A.F.); 2Centro Grandi Strumenti, University of Pavia, 27100 Pavia, Italy; barbara.mannucci@unipv.it; 3Institute of Molecular Genetics, Italian National Research Council (CNR), 27100 Pavia, Italy; leta@igm.cnr.it; 4Department of Molecular Medicine, University of Pavia, 27100 Pavia, Italy; v.rizzo@smatteo.pv.it; 5Fondazione IRCCS Policlinico San Matteo, 27100 Pavia, Italy; 6Emergency Department, Fondazione IRCCS Policlinico San Matteo, 27100 Pavia, Italy

**Keywords:** desaturase, fatty liver, oxidative stress, metalloproteinases, TNF-alpha

## Abstract

We investigated changes in fatty acid desaturases, D5D, D6D, D9-16D and D9-18D, and their relationship with oxidative stress, matrix metalloproteinases (MMPs) and serum TNF-alpha in two rat models of non-alcoholic fatty liver disease NAFLD. Eight-week-old male Wistar rats fed for 3 weeks with methionine-choline–deficient (MCD) diet and eleven-week-old Obese male Zucker rats were used. Serum levels of hepatic enzymes and TNF-alpha were quantified. Hepatic oxidative stress (ROS, TBARS and GSH content) and MMP-2 and MMP-9 (protein expression and activity) were evaluated. Liver fatty acid profiling, performed by GC-MS, was used for the quantification of desaturase activities. Higher D5D and D9-16D were found in Obese Zucker rats as well as an increase in D9-18D in MCD rats. D6D was found only in MCD rats. A negative correlation between D5D and D9-16D versus TBARS, ROS and TNF-alpha and a positive correlation with GSH were shown in fatty livers besides a positive correlation between D9-18D versus TBARS, ROS and TNF-alpha and a negative correlation with GSH. A positive correlation between D5D or D9-16D or D9-18D versus protein expression and the activity of MMP-2 were found. NAFLD animal models showed comparable serum enzymes. These results reinforce and extend findings on the identification of therapeutic targets able to counteract NAFLD disorder.

## 1. Introduction

Non-alcoholic fatty liver disease (NAFLD) is considered the most prevalent form of chronic liver disease in the world due to a global incidence estimated at about 25% [1]. NAFLD is the hepatic manifestation of the metabolic syndrome, a clustering of several interrelated clinical features such as insulin resistance with elevated fasting glycemia, dyslipidaemia, hypertension and visceral obesity. A significant proportion of subjects with NAFLD develops non-alcoholic steatohepatitis (NASH) characterized by hepatocyte damage, inflammation and fibrosis leading to cirrhosis and hepatocellular carcinoma. Understanding the mechanisms involved in NAFLD progression condition thus represents a priority [2].

Fatty livers accumulate lipid droplets when lipid metabolism and export are reduced and lipid influx and synthesis are excessive [3]. Desaturases are lipogenic enzymes that introduce a stereospecific double bond between specific carbons of fatty acyl chains [4]. In mammals, fatty acid desaturases (FADs) are the key enzymes involved in the synthesis of polyunsaturated fatty acids (PUFA) [4]. FADS1 or delta5 desaturase (D5D), introduces a double bond between Carbons 5-6, generally evaluated on 20:3n-6→20:4n6. FADS2 or delta6 desaturase (D6D), introduces a double bond between Carbons 6 and 7, generally evaluated on 18:2n-6 → 18:3n-6.

Stearoyl-CoA desaturase 1 (SCD1) is the final step in *de novo* lipogenesis and converts saturated fatty acids (SFA) to monosaturated fatty acids (MUFAs). SCD1 catalyses the synthesis of MUFAs, primarily oleate and palmitoleate from SFAs, specifically palmitate (D9-16D) and stearate (D9-18D), by introducing a cis double bond between Carbons 9 and 10 of the acyl-CoA substrate [5]. The use of the product-to-precursor ratio of fatty acids has been used as a surrogate of enzyme activity both in animal and human studies [6].

Previous studies documented changes in hepatic desaturases during NAFLD/NASH development: D5D index was found to be lower in genetically obese rats [7] and also a reduction in D6D was detected in high fat diet mice [8]; by contrast, an increase in the hepatic index of SCD1 activity was found in fatty livers and a significant correlation between of SCD1 levels with the severity of hepatic steatosis was documented [9]. However, the role of desaturases has not been completely defined.

Lipotoxicity, arising from the hepatic excess in fat, leads to mitochondrial dysfunction associated with an elevated capacity to oxidize fatty acids (FAs) resulting in the production of reactive oxygen species (ROS). These events cause oxidative stress due to an imbalance between the production of ROS and protective oxidants [10]. Oxidative stress in NAFLD induces activation of hepatic stellate cells (HSCs), the most important producer of extracellular matrix (ECM). A relation between ROS and HSC activation was seen as well as an increase in mRNA expression of Type I collagen and matrix metalloproteinases-2 (MMP-2) through the p38/MAPK signalling pathway [11]. 

Previous experimental and clinical studies performed to counteract NAFLD disorder have shown that dietary supplementation with docosahexaenoic acid (DHA, C22:6) prevents or alleviates NAFLD [12]. 

Animal models of hepatic steatosis have elucidated the mechanisms involved in the pathogenesis of NAFLD. The present study has used two rat models of NAFLD: a nutritional rat model, using a methionine and choline deficient (MCD) diet and a genetic rat model, using Obese Zucker fa/fa rats. To further understand the development and progression of NAFLD, we investigated the changes in fatty acid desaturases and DHA and their relationship with oxidative stress and matrix metalloproteinase activity. Furthermore, the possible associations between desaturase and DHA with serum tumor necrosis factor (TNF)-alpha have been evaluated.

## 2. Materials and Methods 

### 2.1. Materials 

All reagents were obtained from SIGMA (20100 Milano, Italy) and were of the highest grade of purity available. 

### 2.2. Animals

The animal model used was approved by the Italian Ministry of Health and the Pavia University Animal Care Commission (Document number 2/2012). Male Wistar rats (8 weeks old) and Obese (fa/fa) (375 ± 15 g) and Lean (fa/-) (300 ± 10 g) male Zucker rats (11 weeks old) (Charles River, Italy) were used (*n* = 6 each group). Wistar rats were fed with either a methionine-choline deficient diet or a control diet obtained from Piccioni (20060 Gessate, Italy), for 3 weeks. At the time of sacrifice, blood samples and hepatic biopsies from the left lobe were collected and snap frozen in liquid nitrogen.

### 2.3. Serum Measures

Liver injury was assessed by serum levels of alanine transaminase (ALT, mU/mL), aspartate transaminase (AST, mU/mL), alkaline phosphatase (AP, mU/mL), total and direct bilirubin (mg/dl) by an automated Hitachi 747 analyser (Roche/Hitachi, Indianapolis, IN, USA). Serum levels of glucose (mg/dl), p-Cholinesterase (mU/mL), cholesterol (mg/dl) and triglycerides (mg/dl) were also evaluated by an automated Hitachi 747 analyser (Roche/Hitachi, Indianapolis, IN, USA). Inflammation was determined by serum cytokine determination using an Elisa kit for TNF-alpha (pg/mL).

### 2.4. Hepatic Lipid Extraction and Quantification

Hepatic lipid quantification was performed according to Lyn-Cook et al. [13]. Frozen tissues (50–70 mg each) were homogenized in 200 µL of water. Lipids were extracted by adding 1 mL chloroform-methanol (2:1) and samples incubated for 1 h at room temperature with intermittent agitation. After centrifuging at 3000 rpm for 5 min at room temperature, the separated lipid-containing lower fraction was transferred to a clean tube and N_2_-dried. Pellets were re-suspended in 100 µL of 100% ethanol. The fatty acid profile was analysed using a ThermoFisher Scientific DSQII GC/MS system (TraceDSQII mass spectrometer, TraceGCUltra gaschromatograph), Xcalibur MS Software Version 2.1 (including NIST Mass Spectral Library, NIST 08) and Wiley Registry of Mass Spectral Data 8th Edition for assignment of chemical structures to chromatographic peaks. The fatty acid composition was analysed by gas chromatography after being converted to low molecular weight, volatile, nonpolar derivatives, such as fatty acid methyl esters. This conversion uses a transesterification—the alcohol portion of the molecule is displaced by another alcohol, in the presence of an acidic catalyst (e.g., hydrochloric acid), which can be removed, along with excess alcohol, when the reaction is completed. Methanolic HCl 2 N was used to derivatize the samples, converting the fatty acids into methylated fatty acids (FAMEs). In our study 5 µL aliquots of rat liver extracts were dissolved in 1 mL methanolic HCl 2 N into reaction vials. The vials were capped and heated at 70–80 °C for 4 h. The samples were allowed to cool, then dried under a nitrogen stream. Later, 250 µL of dichloromethane was added and 1 µL aliquot was sampled for analysis. Dichloromethane was used as a blank to avoid carryover from previous analysis. The reference standard Marine Oil FAME Mix from Restek (cat. 35066) was used to identify and quantify the fatty acids. The multianalyte standard solution was 10–160 µg/mL in hexane. The hepatic fatty acid profile was expressed in nmol/g liver.

Desaturase activity can be approximated by calculating the product-to-precursor ratio of fatty acids [14]. Desaturase activity indices were calculated as follows: D5, 20:4 n = 6/20:3 n = 6; D6, 18:3 n = 6/18:2 n = 6; D9-16, 16:1 n = 7/16:0; D9-18, 18:1 n = 9/18:0 [15]. 

### 2.5. Protein Levels and Activity of Matrix Metalloproteinases (MMP-2, MMP-9)

The protein content of MMP-2 and MMP-9 were quantified by Western Blot assay (MMPs/tub). CelLytic Buffer and Protease Inhibitor Cocktail were purchased from Sigma-Aldrich (Milan, Italy), as well as the mAb anti-alpha-tubulin (DM1A). Rabbit polyclonal antibodies anti-MMP-2 and anti-MMP-9 were obtained from Thermo-Scientific (USA). Liver tissue samples were homogenized in ice-cold Cell Lytic Buffer supplemented with Protease Inhibitor Cocktail and centrifuged at 15,000 g for 10 min. The collected supernatant was divided into aliquots containing the same amount of proteins and stocked at −80 °C. Samples of liver extracts containing the same amount of proteins were separated in SDS-PAGE on 7.5% acrylamide gels and transferred to the PVDF membranes. Aspecific sites were blocked for 2 h with 5% Bovine Serum Albumin (BSA) in TBS (20 mM Tris/HCl, 500 mM NaCl, pH 7.5, 0.1% Tween 20) at 4 °C. The membranes were incubated with primary antibodies overnight at 4 °C under gentle agitation. Primary antibodies against alpha-tubulin, MMP-2, MMP-9 were used at a dilution of 1:1000. Membranes were washed in PBS (Na_2_HPO_4_ 8 mM, NaH_2_PO_4_-H_2_O 2 mM, NaCl 140 mM, pH 7.4, 0.1% Tween 20) and incubated with peroxidase-conjugated secondary antibody, at a 1:2000 dilution. Immunostaining was revealed with BIO-RAD Chemidoc XRS+. Bands intensity quantification was performed by BIO-RAD Image Lab software (20090 Segrate, Italy).

In order to detect MMPs activity (OD/mg/mL prot) in the samples, the homogenate protein content was normalized by a final concentration of 400 µg/mL in a sample loading buffer (0.25 M Tris-HCl, 4% sucrose w/v, 10% SDS w/v and 0.1% bromphenol blue *w/v*, pH 6.8). After dilution the samples were loaded onto electrophoretic gels (SDS-PAGE) containing 1 mg/mL of gelatin under non reducing conditions [16] followed by zymography as described previously [17]. The zymograms were analysed by densitometer (GS 900 Densitomer BIORAD, Hercules, CA, USA) and data were expressed as optical density (OD) reported as OD/ mg/mL protein.

### 2.6. Oxidative Stress

The hepatic concentration of total glutathione (GSH, nmol/mg prot)) was measured by an enzymatic method (Cayman Chemical Co., Ann Arbor, MI, USA). The extent of liver lipid peroxidation in terms of thiobarbituric acid reactive substances (TBARS, nmol/mg prot) formation was measured according to the method of Esterbauer and Cheeseman [18]. The TBARS concentrations were calculated using malondialdehyde (MDA) as standard. ROS (A.U.) were quantified by the DCFH-DA method based on the ROS-dependent oxidation of DCFH to DCF, as already described in detail [19]. Protein content was assayed by the method of Lowry et al. [20].

### 2.7. Statistical Analysis

Normal data distribution were analysed by one-way ANOVA, followed by Tukey’s multiple comparisons test. When data distribution was not normal according to the Kolmogorov-Smirnov test, the Mann-Witney test was used. The correlation analysis was defined according to Pearson correlation (r) or Spearman (rs). The value of *p* < 0.05 was considered to indicate statistical significance. The accompanying table and graphs present the mean value ± standard error of the mean (SEM). Statistical analysis was performed using MedCalc Statistical Software version 18.11.3 (Eron S.r.l., 80013 Casalnuovo di Napoli, Italy).

## 3. Results

### 3.1. Desaturase Activity Index in MCD e Zucker Rats

The evaluation of D5D (20:4 n-6/20:3 n-6), D6D (18:3 n-6/18:2 n-6), D9-16D (16:1 n-7/16:0), D9-18D (18:1 n-9/18:0) activity indexes in fatty livers obtained from MCD e Zucker rats is reported in Table 1. Higher D5D and D9-16D were found in Obese Zucker as compared with MCD rats. On the contrary, a marked increase in D9-18D occurs in MCD rats as compared with Obese Zucker rats. D6D was found only in MCD rats and increased when compared with its respective control rats. Lower D5D and higher D9-18D were found in fatty livers when compared with their respective controls. D9-16D increased in Obese Zucker rats when compared with Lean Zucker rats (Table 1).

The NAFLD rat models used differ as regards oxidative stress: in particular, a hepatic increase in ROS and TBARS and a decrease in GSH content were found in MCD rats (Table 1). The evaluation of protein content and activity of MMPs showed higher levels of MMP-2 in MCD versus Obese Zucker rats (Table 1). Protein expression and activity of MMP-9 were detectable only in MCD rats and increased when compared with their control rats (Table 1).

### 3.2. Correlation/Association between Desaturase Activity Indexes and DHA Versus Oxidative Stress 

Using the hepatic fraction of fatty livers from MCD and Obese Zucker rats, the activity index of the desaturases was compared with oxidative stress. A negative correlation between D5D versus ROS and TBARS supported by a positive correlation with GSH was found (Figure 1, panel a-c). A positive correlation between D6D versus TBARS and GSH was found in MCD rats (Figure 1 panel d-f).

A negative correlation between D9-16D versus ROS and TBARS supported by a positive correlation with GSH was shown (Figure 2, panel a–c). On the contrary, a positive correlation between D9-18D versus ROS and TBARS supported by a negative correlation with GSH was shown (Figure 2, panel d–f). 

A negative correlation between DHA and ROS and TBARS supported by a positive correlation with GSH was detected (Figure 3).

### 3.3. Correlation between Desaturase Activity Indexes and DHA Versus Matrix Metalloproteinases

A positive correlation between D5D and D9-16D and D9-18D versus activity and protein expression of MMP-2 was found in livers from MCD and Obese Zucker rats (Figure 4 and Figure 5). In MCD rats a good correlation between D9-18D versus protein content of MMP-9 was detected (r_s_ 0.99, *p* < 0.0001).

A negative correlation between DHA versus MMP-2 activity was found (Figure 6, panel a). In addition, the same trend was detected for MMP-9 activity in MCD rats (Figure 6, panel b).

### 3.4. Correlation between Serum Levels of TNF-alpha Versus Liver Desaturases, DHA and MMPs

In livers from MCD and Obese Zucker rats, a negative correlation between D5D or D9-16D versus TNF-alpha was found (Table 2). The same trend occurred between DHA versus TNF-alpha (Table 2). On the contrary, a strong positive correlation between D9-18D versus TNF-alpha was detected as well as for MMP-2 (activity and protein) versus TNF-alpha (Table 2). 

### 3.5. Dietary and Genetic Model of NAFLD

Dietary (MCD) and genetic (Obese Zucker) models features are summarized in Table 3. Comparable serum levels of ALT, AST and Alkaline phosphatase are shown. Serum differences were found for total and direct bilirubin and p-Cholinesterase; an approximately ten-fold increase in cholesterol and triglycerides was found in Obese Zucker versus MCD rats. The serum levels of cholesterol and triglyceride were comparable with those reported previously [21,22]. A marked higher levels of TNF-alpha were detectable in MCD when compared with Obese Zucker rats. 

### 3.6. Hepatic Fatty Acid Profiles

Hepatic fatty acid profiles in MCD and Zucker rats were reported in Table 4. MCD diet induced a significant increase (*p* < 0.05) in Total SFA, Total MUFA and Total PUFA in comparison with Obese Zucker rats. Moreover, MCD rats showed a decreasing trend of Total SFA, MUFA and PUFA in comparison with control groups, reaching significance with regard to the Total SFA. Ratios of SFA/MUFA and PUFA/MUFA were significantly lower (*p* < 0.05) in MCD when compared to Obese Zucker rats.

## 4. Discussion

The current study documented significant correlations between desaturases and DHA with oxidative stress as well as with MMPs activity in livers from MCD and Obese Zucker rats. In addition, significant associations were found comparing desaturases, DHA and MMPs with serum TNF-alpha levels.

### 4.1. Desaturases and DHA Versus Oxidative Stress

A central role in the pathogenesis of NAFLD is assigned to oxidative stress since the onset of fatty liver accumulation is associated with changes in the hepatocytes redox status [23].

A decrease in desaturase activity, D5D and D6D, was detected in livers obtained from mice or rats submitted to high-fat diet (HFD) [15,24]. The same trend occurred in NAFLD patients [25]. A recent study, using livers of high-fat diet (HFD) mice, documented a correlation of D5D and D6D with oxidative stress–related indexes such as GSH/GSSG and TBARS [8]. The same trend was also found for D5D in the current work, using other rat models of fatty livers. These correlations showed that D5D is privileged under low oxidative stress conditions either through low levels in ROS and TBARS and high GSH content. Opposite results were obtained for D6D, the latter detectable only in MCD rats, probably due to the difference in the diet used. Of note, the same trend in D5D and D6D that we documented were found in plasma of patients with NAFLD compared with control: lower levels in D5D and higher levels in D6D [26].

The correlation between D9-16D and D9-18D with oxidative stress documented two different trends: D9-16D negatively correlated with oxidative stress, on the contrary, an opposite trend was documented for D9-18D. The expression of SCD1 gene, for D9-16D and D9-18D, is associated with the degree of steatosis [27]. A link between SCD1, inflammation and NASH has been demonstrated [28]. Previous studies have shown that SCD1 enzyme plays a key role in lipid partitioning in the liver [29]—hence the inhibition of SCD1 reduced hepatic steatosis and inflammation [30]. A loss of SCD1 reduces adipocyte inflammation and its paracrine control of inflammation in macrophages and endothelial cells [31]. In several studies the distinction between D9-16D and D9-18D is absent [30]. In the present study we evaluated either D9-16D and D9-18D demonstrating that D9-16 is negatively associated with TBARS, ROS and TNF-alpha and positively associated with GSH. On the contrary D9-18 was positively associated with TBARS, ROS and TNF-alpha and positively associated with GSH. Thus D9-16D and D9-18D seem to be involved in opposite effects in the experimental rat models considered in this study. Our experiments reinforced data showing a positive correlation between D9-18D and serum levels of TNF-alpha demonstrating the inhibition of the D9-18D pathway which may be considered as a pathway controlling inflammation.

DHA exhibits antioxidant and anti-inflammatory effects in MCD diet-induced NAFLD [32]. In addition, a diet rich in DHA improved lipid metabolism and showed anti-inflammatory effects in HFD-induced NALFD in C57BL/6J mice [33] as well as the administration of DHA was able to block progression of Western diet-induced NASH [34]. Our data demonstrate a significant negative correlation between DHA with TBARS and serum levels of TNF-alpha, the latter a key factor in the development of NAFLD [35]. On the other hand, the oxidative stress, in particular ROS and lipid peroxidation products, is directly involved in the Kupffer cell induction that produces TNF-alpha [36]. To further support the antioxidant role of DHA, we further demonstrated that a marked decrease in DHA occurred especially in MCD in which a considerable oxidative stress occurred.

### 4.2. Desaturase and DHA versus MMPs

The progression of NAFLD to NASH is characterized by the involvement of ECM remodelling, which is a complex and dynamic event, still not completely elucidated. MMPs play a crucial role in ECM homeostasis due to HSC activation into a profibrogenic phenotype occurring during inflammation [37,38]. Recent data have shown an ongoing remodelling activity of MMP-2 in advanced human NAFLD fibrosis [39]. In the present study, we documented, for the first time, a positive correlation between desaturases, D5D, D9-16D and D19-18D, with MMP-2 protein content and its gelatinolytic activity in NAFLD animal models. It has been shown that high levels of PUFAs and a very high n-6/n-3 ratio may contribute to the increased incidence of cancer via upregulation of matrix metalloproteinase-1 (MMP-1) [40] and that the decreased n-6/n-3 fatty acid ratio downregulates the expression of MMP-1 [41]. Our findings suggest that desaturases also play a role in the progression of NAFLD via MMPs. PUFAs have been reported to play a protective role in a wide range of diseases characterized by increased MMPs activity [42,43,44,45,46]

The inhibition exerted by DHA, a compound with six double bonds in the structure, is comparable to that of the α-linolenic and γ-linolenic acids. The negative correlation between DHA and MMP-2 and MMP-9 activities, reported in the current study, is strongly supported by previous studies [47,48] asserting that chain unsaturation and a 18–20 carbon chain length are two factors favouring the inhibition of MMP activity by fatty acids. Preservation of the collagen matrix integrity is an important issue for improving and reducing NAFLD progression. Progression of NAFLD to NASH is also modulated by TNF-alpha. It is known that SCD1-deficiency attenuates the induction of TNF-alpha [49], involved in the control of MMPs [50]; in addition, silencing of TNF-alpha in myeloid cells is able to prevent the development of NASH [51]. The current study supports a strong association between desaturases, TNF-alpha and MMP-2 in rat models of NAFLD. Moreover, the strong positive correlation between protein content and MMP-2 activity index with serum levels of TNF-alpha, shown in our data, reinforces previous results about the association between ECM remodelling and inflammation. No significant correlation was found for MMP-9 probably due to the low sample size associated to its identification only in MCD and absent in Obese Zucker rats.

Previous studies reported design, synthesis and biological evaluation of desaturase inhibitors, especially for SCD1. Although SCD1 is ubiquitously expressed, it is predominant in lipogenic tissues, especially hepatocytes and adipocytes [52]. Recent data obtained both in mice and rats have reported the discovery of potent SCD1 inhibitors able to reduce steatosis without significant adverse events thus representing innovative and promising therapeutic compounds for NAFLD treatment [53].

## 5. Conclusions

In conclusion, using rat livers from MCD and Obese Zucker rats, this study has documented (i) the changes on hepatic D5D, D6D, D9-16D, D9-18D; (ii) a correlation between desaturases and DHA with oxidative stress; (iii) a correlation between desaturases and DHA with MMPs proteolytic activity; (iv) a correlation between serum levels of TNF-alpha with hepatic desaturases and DHA.

Although much research remains to be done to clarify the pathophysiology of NAFLD, these results reinforce and extend findings on the identification of potential therapeutic targets able to counteract this common disorder.

## Figures and Tables

**Figure 1 nutrients-11-00799-f001:**
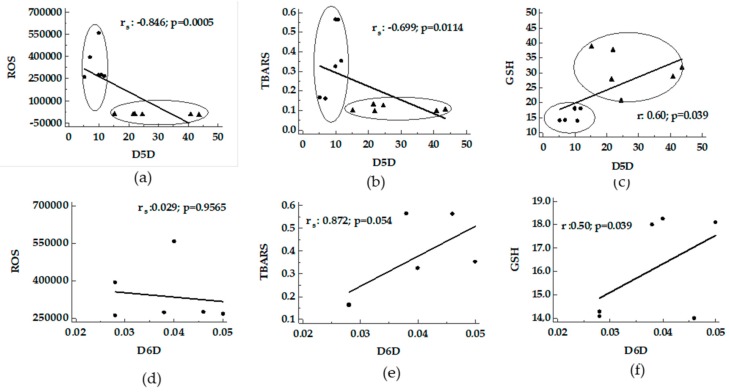
Correlation between desaturase activity indexes and oxidative stress in livers from MCD and Obese Zucker rats. Panels (**a**–**c**), D5D activity index versus ROS, TBARS and GSH; panels (**d**–**f**), D6D activity index, detected only in MCD rats, versus ROS, TBARS and GSH. ●: MCD rats; ▲: Obese Zucker rats. TBARS, thiobarbituric acid reactive substances; ROS, reactive oxygen species; GSH, glutathione.

**Figure 2 nutrients-11-00799-f002:**
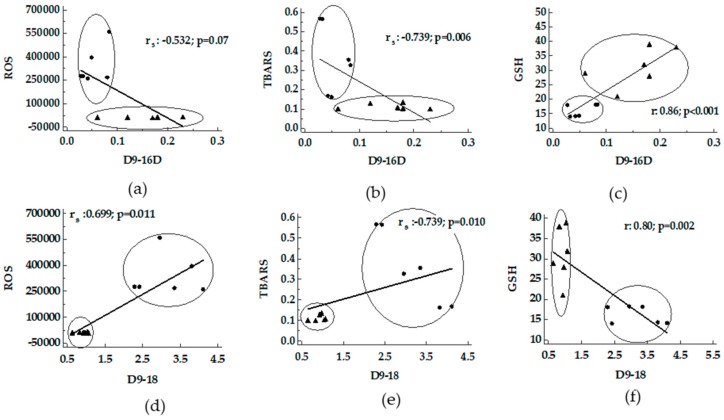
Correlation between desaturase activity indexes and oxidative stress in livers from MCD and Obese Zucker rats. Panels (**a**–**c**), D9-16D activity index versus ROS, TBARS and GSH; panels (**d**–**f**), D9-18D activity index versus ROS, TBARS and GSH. ●: MCD rats; ▲: Obese Zucker rats. TBARS, thiobarbituric acid reactive substances; ROS, reactive oxygen species; GSH, glutathione

**Figure 3 nutrients-11-00799-f003:**
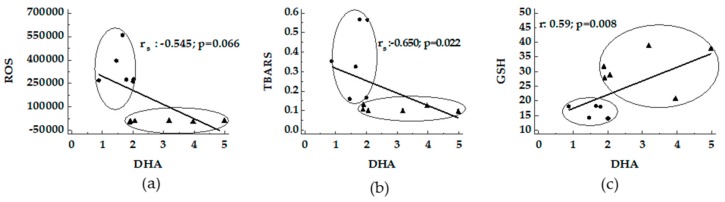
Correlation between DHA and oxidative stress in livers from MCD and Obese Zucker rats. The association between DHA versus ROS (panel **a**), TBARS (panel **b**) and GSH (panel **c**) were reported. ●: MCD rats; ▲: Obese Zucker rats.

**Figure 4 nutrients-11-00799-f004:**
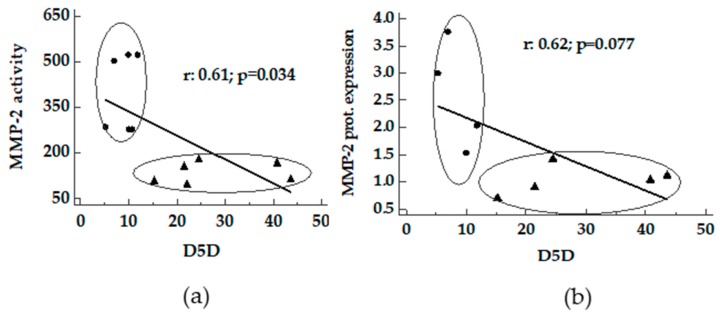
Correlation between D5D and MMPs in livers from MCD and Obese Zucker rats. Panels **a** and **b**, D5D activity index versus MMP-2 activity and MMP-2 protein expression. ●: MCD rats; ▲: Obese Zucker rats.

**Figure 5 nutrients-11-00799-f005:**
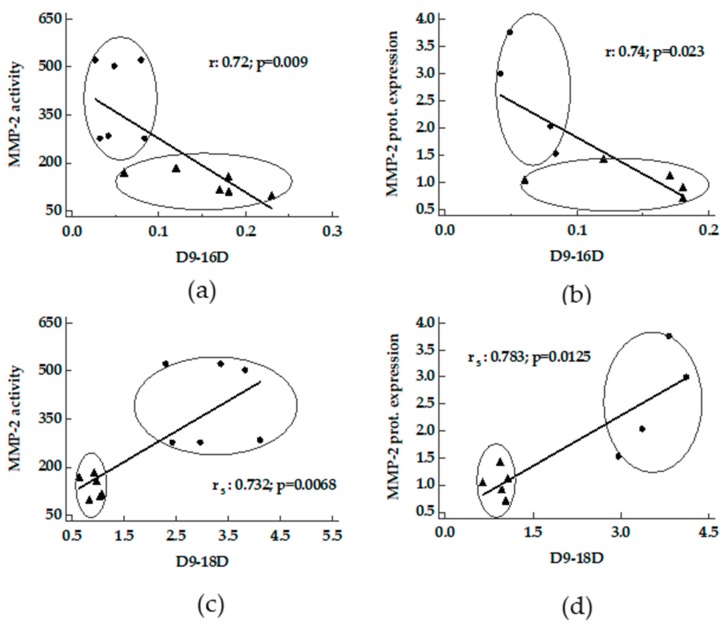
Correlation between desaturases and MMPs in livers from MCD and Obese Zucker rats. Panels **a** and **b**, D9-16D activity index versus MMP-2 activity and MMP-2 protein expression; panels **c** and **d**, D9-18D activity index versus MMP-2 activity and MMP-2 protein expression. ●: MCD rats; ▲: Obese Zucker rats.

**Figure 6 nutrients-11-00799-f006:**
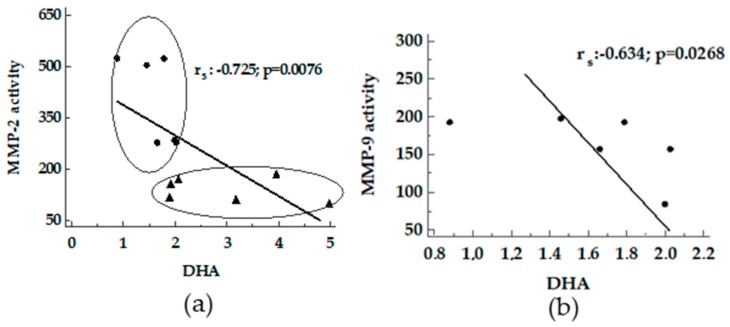
Correlation between DHA and MMPs in livers from MCD and Obese Zucker rats. Panels **a**, DHA versus MMP-2 activity; panels **b**, DHA versus MMP-9 activity. ●: MCD rats; ▲: Obese Zucker rats.

**Table 1 nutrients-11-00799-t001:** Hepatic desaturases, oxidative stress and matrix metalloproteinases (MMPs) in methionine-choline-deficient (MCD) and Obese Zucker rats.

	Control MCD	MCD	*p* ^1^	Lean Zucker	Obese Zucker	*p* ^1^	*p* ^1^
D5D	37.06 ± 4.3	8.89 ± 0.9	<0.001	56.65 ± 12.5	27.09 ± 4.0	<0.001	**#**
D6D	0.0076 ± 0.0012	0.0383 ± 0.0037	<0.001	n.d.	n.d.	n.d.	**n.d.**
D9-16D	0.0565 ± 0.0082	0.0510 ± 0.0084	<0.001	0.040 ± 0.0049	0.153 ± 0.021	<0.001	**#**
D9-18D	1.249 ± 0.177	3.266 ± 0.273	<0.001	0.437 ± 0.055	0.909 ± 0.054	<0.001	**#**
TBARS	0.06 ± 0.003	0.34 ± 0.07	<0.05	0.14 ± 0.01	0.11 ± 0.0006	<0.05	**§**
ROS	26,677.3 ± 7691.3	3,384,750.2 ± 4844.4	<0.05	27,671.0 ± 10,713.9	12,498.2 ± 987.4	ns	**§**
GSH	37.5 ± 1.6	16.1 ± 0.9	<0.05	35.7 ± 2.0	31.2 ± 2.7	ns	**§**
MMP-2 activity	227.3 ± 17.3	397.1 ± 53.0	<0.05	192.0 ± 5.8	139.7 ± 14.3	ns	**§**
MMP-9 activity	92.8 ± 12.5	163.3 ± 17.6	<0.05	n.d.	n.d.	ns	**n.d.**
MMP-2 protein	1.8 ± 0.3	2.6 ± 0.5	ns	1.3 ± 0.2	1.1 ± 0.1	ns	**§**
MMP-9 protein	0.2 ± 0.007	0.5 ± 0.1	ns	n.d.	n.d.	ns	**n.d.**

^1^ ns: *p* value not significant; #: *p* < 0.001 or §: *p* < 0.05 MCD vs. Obese; n.d.: not detectable. TBARS, thiobarbituric acid reactive substances; ROS, reactive oxygen species; GSH, glutathione.

**Table 2 nutrients-11-00799-t002:** Correlation between serum levels of TNF-alpha versus liver desaturase, DHA and MMPs.

	Serum TNF-alpha	
	r/r_s_	*p*
D5	−0.708	0.010
D6	−0.040	ns
D9-16	−0.611	0.035
D9-18	0.830	0.0008
DHA	−0.802	0.0017
MMP-2 activity	0.909	<0.0001
MMP-2 protein	0.887	0.0014

Pearson correlation (r) or Spearman (rs); ns: *p* value not significant. DHA, docosahexaenoic acid.

**Table 3 nutrients-11-00799-t003:** Serum parameters in MCD and Obese Zucker rats.

	Control MCD	MCD	*p* ^1^	Lean Zucker	Obese Zucker	*p* ^1^	*p* ^1^
ALT	23.3 ± 1.6	124.0 ± 34.0	<0.05	68.0 ± 3.5	109.3 ± 16.9	<0.05	**ns**
AST	59.5 ± 4.8	128.0 ± 16.8	<0.05	114.7 ± 18.0	111.5 ± 8.9	ns	**ns**
Alkaline phosphatase	134.0 ± 7.3	186.3 ± 25.3	ns	191.0 ± 5.1	194.2 ± 10.7	ns	**ns**
Glucose	179.5 ± 11.7	151.2 ± 7.5	<0.05	128.3 ± 10.7	144.2 ± 8.4	ns	**ns**
Total bilirubin	0.11 ± 0.004	0.26 ± 0.02	<0.05	0.11 ± 0.001	0.13 ± 0.002	ns	**§**
Direct bilirubin	0.12 ± 0.002	0.14 ± 0.02	<0.05	0.1 ± 0.001	0.12 ± 0.002	ns	**§**
p-Cholinesterase	272.4 ± 10.9	886.2 ± 41.4	<0.05	478.7 ± 39.9	651.4 ± 28.7	<0.05	**§**
Cholesterol	64.7 ± 4.1	23.8 ± 1.6	<0.05	100.0 ± 3.5	180.5 ± 4.8	<0.05	**§**
Triglycerides	72.3 ± 16.8	16.8 ± 2.3	<0.05	40.0 ± 5.1	215.5 ± 38.2	<0.05	**§**
TNF-alpha	27.2 ± 1.1	36.9 ± 2.3	<0.05	9.7 ± 0.5	9.9 ± 0.4	ns	**§**

^1^ ns: *p* value not significant; §: *p* < 0.05 MCD vs. Obese. ALT, alanine transaminase; AST, aspartate transaminase; TNF-alpha, tumor necrosis factor-alpha.

**Table 4 nutrients-11-00799-t004:** Hepatic fatty acid composition in MCD and Obese Zucker rats.

	ControlMCD	MCD	*p* ^1^	Lean Zucker	Obese Zucker	*p* ^1^	*p* ^1^
16:00	66.44 ± 12.5	32.64 ± 5.13	<0.05	53.52 ± 16.27	11.05 ± 2.58	<0.05	**§**
18:00	27.25 ± 4.45	8.48 ± 0.89	<0.001	27.40 ± 5.22	8.30 ± 2.12	<0.001	**ns**
**Total SFA**	93.69 ± 15.07	41.13 ± 5.91	<0.05	80.97 ± 21.90	19.36 ± 4.67	<0.05	**§**
16:1 n-7	3.2 ± 0.35	1.59 ± 0.26	<0.05	2.39 ± 0.87	1.94 ± 0.69	ns	**§**
18:1 n-7	4.44 ± 0.83	2.93 ± 0.49	ns	4.76 ± 1.0	1.57 ± 0.48	<0.05	**ns**
18:1 n-9	31.6 ± 4.99	28.2 ± 4.61	ns	13.3 ± 3.37	7.4 ± 1.70	ns	**§**
**Total MUFA**	39.25 ± 5.46	32.72 ± 5.28	ns	20.43 ± 4.93	10.63 ± 2.93	ns	**§**
18:2 n-6	52.1 ± 7.67	63.88 ± 9.75	ns	17.78 ± 3.9	2.75 ± 0.66	<0.05	**§**
18:3 n-6	0.41 ± 0.09	2.33 ± 0.34	<0.05	0.38 ± 0.07	n.d.	n.d.	**n.d.**
20:3 n-6	0.66 ± 0.16	1.34 ± 0.30	ns	0.60 ± 0.13	0.40 ± 0.10	ns	**§**
20:4 n-6	45.17 ± 11.97	11.76 ± 1.15	<0.05	66.19 ± 19.22	9.58 ± 1.49	<0.05	**ns**
22:5 n-3	1.76 ± 0.57	1.71 ± 0.14	ns	11.90 ± 5.4	0.67 ± 0.08	<0.05	**§**
22:6 n-3	2.80 ± 0.43	1.83 ± 0.24	<0.05	6.57 ± 1.04	3.2 ± 0.49	<0.05	**§**
**Total PUFA**	102.16 ± 19.25	82.55 ± 11.51	ns	102.02 ± 29.79	16.54 ± 2.43	<0.05	**§**
**SFA/MUFA**	2.37 ± 0.17	1.28 ± 0.036	<0.05	4.17 ± 0.40	1.96 ± 0.14	<0.05	**§**
**PUFA/MUFA**	2.56 ± 0.25	2.63 ± 0.19	ns	5.99 ± 1.35	1.85 ± 0.25	<0.05	**§**

^1^ ns: *p* value not significant; §: *p* < 0.05 MCD vs. Obese; n.d.: not detectable. SFA, saturated fatty acids; MUFA, monosaturated fatty acids; PUFA, polyunsaturated fatty acids.

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
