# Peer review of "Fatty Acid Desaturase Involvement in Non-Alcoholic Fatty Liver Disease Rat Models: Oxidative Stress Versus Metalloproteinases"

_nutrients, 2019, doi:10.3390/nu11040799_

Reviewer 1 Report

The present study aims to determine the changes in lipid desaturase activities in two models of fatty liver in rats and investigate its potential relevance for liver disease, highlighting relevant links with ROS control and fibrosis mediators.

The mayor caveat of the study is the style that could be greatly improved by a native English speaker to highlight the main findings and significance.

Other than that just a few minor suggestions. In fig. 1 the main comparison should be between MCD and their controls and Lean vs Obese, so the order in the graphs should be altered, also multiple comparison statistical analysis should be included. for fig. 2, 3, and 4 the groupings of the MCD and obese data points would increase the clarity of the results and could be done simply. In the corresponding text rather than "fatty livers" it should be specified that this refers to MCD and obese livers. Finally, data for the control mice should be included at least in a supp. figure.

Author Response

We thank the referee for the useful suggestions that have improved our manuscript.

1) The mayor caveat of the study is the style that could be greatly improved by a native English speaker to highlight the main findings and significance.

Reply: The manuscript  has been revised by Prof. Anthony Baldry, full Professor in English Linguistics, and former Associate Professor of English at the Faculty of Medicine, the University of Pavia. Prof. Baldry has read through, and revised, the English phraseology and terminology in keeping with the standards required by native speakers of English. We are available to provide the certificate.

2) In fig. 1 the main comparison should be between MCD and their controls and Lean vs Obese, so the order in the graphs should be altered, also multiple comparison statistical analysis should be included.

Reply: To further underline the comparison between MCD and their controls and Lean vs Obese, the figure 1 has been changed in a new table and as suggested the order of the columns have been modified.

For figures 2, 3, and 4 the groupings of the MCD and obese data points would increase the clarity of the results and could be done simply.

Reply: The groupings of the MCD and obese data points have been added.

In the corresponding text rather than "fatty livers" it should be specified that this refers to MCD and obese livers. Finally, data for the control mice should be included at least in a supp. figure.

Reply: “fatty liver” has been changed in MCD and obese livers. Data from the control rats have been added in table 1.

Reviewer 2 Report

The authors investigated the involvement of fatty acid desaturase in non-alcoholic fatty liver disease (NAFLD) rat model. The fatty acid desaturases D5D, D6D, D9-16D and D9-18D showed different expression in different model of rats and correlated with oxidative stress related markers. While the contents are interesting, there are several points that still need to be addressed.

Major

1. In Figure 2 to Figure 7; The level of fatty acid desaturases and oxidative stress related markers of different rat models were plotted in mixed manner. However, it is confusing that the plots of different models were shown in one graph. This should be changed probably to tables showing each markers median or average. Example table is shown.                                

MCDObese
D5D1030
ROS3000010

2. In Figure 1 and in another tables, the order of the group is strange. As the “control MCD” is the complete control group, this should be placed in the first lane. The “lean Zucker” might be in the second, “obese Zucker” in the third, and “MCD” in the last.

Author Response

We thank the referee for the useful suggestions that have improved our manuscript.

1) In Figure 2 to Figure 7; The level of fatty acid desaturases and oxidative stress related markers of different rat models were plotted in mixed manner. However, it is confusing that the plots of different models were shown in one graph. This should be changed probably to tables showing each markers median or average.

Reply: As suggested, in the revised version of the manuscript,  we have added the desaturase average values to a new table. To increase the clarity of the results we have also grouped the MCD and obese data points (figure 2 to figure 7).  As reported in clinical studies, we have used the results of these two NAFLD models together to reproduce the human pathology.  The NAFLD rat models used have been selected with comparable serum levels of ALT, AST, Alkaline phosphatase and glucose.  In addition, in the revised manuscript, “fatty livers" has been specified that refers to MCD and obese livers.

2) In Figure 1 and in another tables, the order of the group is strange. As the “control MCD” is the complete control group, this should be placed in the first lane. The “lean Zucker” might be in the second, “obese Zucker” in the third, and “MCD” in the last.

Reply: As suggested we have changed the order of the groups underlining  the two animal models with their respective control.

Round  2

Reviewer 2 Report

The authors responded as requested.

Author Response

The manuscript  has been revised by Prof. Anthony Baldry, full Professor in English Linguistics, and former Associate Professor of English at the Faculty of Medicine, the University of Pavia. Prof. Baldry has read through, and revised, the English phraseology and terminology in keeping with the standards required by native speakers of English.